# Effect of Ultrasonic and Microwave Dual-Treatment on the Physicochemical Properties of Chestnut Starch

**DOI:** 10.3390/polym12081718

**Published:** 2020-07-31

**Authors:** Meng Wang, Yanwen Wu, Yongguo Liu, Jie Ouyang

**Affiliations:** 1Beijing Key Laboratory of Forest Food Process and Safety, Department of Food Science and Engineering, College of Biological Sciences and Technology, Beijing Forestry University, Beijing 100083, China; wmm666@bjfu.edu.cn; 2Beijing Center for Physical and Chemical Analysis, Beijing Food Safety Analysis and Testing Engineering Research Center, Beijing Academy of Science and Technology, Beijing 100089, China; wuyanwen@bcpca.ac.cn; 3Beijing Key Laboratory of Flavor Chemistry, Beijing Technology and Business University, Beijing 100048, China; liuyg@th.btbu.edu.cn

**Keywords:** starch, ultrasound, microwave, dual treatment, long- and short-range molecular order

## Abstract

This work examined the effect of ultrasound and microwave treatments, separate and in combination, on the physicochemical and functional properties of chestnut starch. The results revealed that the ultrasonic-microwave (UM) and microwave-ultrasonic (MU) dually modified samples exhibited more severe surface damage, weaker birefringence, and lower relative crystallinity and gelatinization enthalpy than the native and single-treated starches. The UM samples showed the highest oil absorption capacity, and the MU samples showed the highest water absorption capacity and the best freeze-thaw stability (five cycles) among all samples. The swelling power, peak, trough, final, and breakdown viscosities, and pasting temperature all decreased regardless of single or dual modification. This study provides a reference for potential industrial applications of ultrasound and microwave treatments for the modification of chestnut starch.

## 1. Introduction

Chestnut is an important woody grain in many parts of the world, including Asia, Europe, and North America, and China is the largest producer [1]. The annual production of Chinese chestnut is approximately 2,000,000 tons (Food and Agriculture Organization (FAO) Statistics) and about 30% of the fruit deteriorates due to mildew, hardening, and germination. Therefore, it is necessary to process chestnut timely after harvest. Starch accounts for approximately 50% of the dry weight of chestnut which has good commercial prospects [2]. Previous studies have shown that chestnut starch presented a lower swelling power and gelatinization parameters (T_o_, T_p_, and T_c_) than waxy corn starch [3,4,5,6]. Liu et al. [6] demonstrated that native chestnut starch gel showed higher texture values and lower syneresis compared to native corn starch. Starch is a low-cost raw material that is used in both food and non-food applications. However, some of the inherent properties of native starch, such as its insolubility in cold water and the poor stability of starch paste, have limited its wider application in food and industrial production [3]. Starch is usually modified by various strategies including physical, chemical, and enzymatic methods to meet the needs of industrial applications.

The development of improved strategies for the physical modification of starch has attracted considerable interests from researchers and consumers particularly due to the safe, non-polluting, simple, and eco-friendly of mechanical processes [7]. Ultrasound and microwave treatments are new physical technologies that can be used to modify the structural properties of materials. Ultrasonic treatment can produce a cavitation effect in the starch–water system that induces micro-jets, shear forces, free radical interactions with starch components, and localized heating, resulting in structural changes in the starch that alter its physicochemical and functional properties [8]. Microwaves are non-ionizing electromagnetic waves that cause micro-motion and friction of molecules in foods by high-frequency electric fields according to dielectric loss effect. Microwave treatment can destroy glycosidic bonds, induce the rearrangement of starch molecules, and affect the morphology of starch granules and relative crystallinity, resulting in changes in pasting viscosity, swelling capacity, emulsifying stability, and emulsifying activity [9,10].

Several studies have examined the individual effects of ultrasound or microwave treatments on starch, such as potato starch, waxy corn starch, and lotus seed starch, but very limited studies have focused on chestnut starch. In addition, the dual physical modification, also referred to as the synergistic effect of physical modification, is very promising because it can improve the properties of starch without involving the use of chemical reagents. The literature included dual modification by annealing and ultra-sonication treatments on potato starch [11], and heat-moisture combined with high pressure treatment of foxtail millet starch [12], etc. Previous reports also pointed out that dual physical modification is more effective than a single treatment, e.g., potato starch had a higher oil absorption capacity after a combination of ultrasound and freeze-thaw cycle treatment than ultrasonic or freeze-thaw cycle treatment alone [13]. To the best of our knowledge, there are few studies on the simultaneous effects of ultrasound and microwave modification on chestnut starch. Therefore, the present study was designed to investigate: (i) the effect of ultrasonic and microwave dual modification on the structural characteristics, thermal and pasting behavior, and functional properties of chestnut starch; (ii) the difference caused by the sequence of using microwave and ultrasound treatments. The results of this research will stimulate further interest in the use of combined physical treatments to improve the properties of starch-based products.

## 2. Materials and Methods

### 2.1. Raw Materials

Chinese chestnut (*Castanea mollissima*) fruits “Dabanhong” were obtained from Kuancheng, Hebei Province of China. Chestnut starch was isolated by alkaline solution according to the method of our previous report [14], and the total starch content in the purified chestnut starch was 91.33%, and the amylose ratio was 31.12%. All chemical reagents were of analytical grade and purchased from Beijing Kebaiao Biotechnology Co., Ltd. (Beijing, China).

### 2.2. Preparation of Samples

#### 2.2.1. Ultrasonic Sample Preparation

Native chestnut starch (2 g) was suspended in distilled water (10%, *w*/*v*). The suspension was treated using an ultrasonic breaker (Biosafer 650-92, Saifei (China) Co., Ltd., Shanghai, China) with a power of 500 W and a frequency of 20 kHz for 60 min (pulse 2 s, intermittent 2 s) [8]. The sonication probe (diameter 6 mm) was immersed in the starch suspension. The starch sample was placed in an ice bath to keep the temperature below 25 °C during the treatment. The treated starch samples were dried at 40 °C for 24 h to obtain ultrasonic modified starch (UC).

#### 2.2.2. Microwave Sample Preparation

For microwave modification, the chestnut starch (2 g) was dispersed in 20 mL distilled water, and the suspension was treated in a microwave oven (MG08S-2B, Nanjing Huiyan Microwave System Co., Ltd., Nanjing, China) at 2450 MHz and 30 W for 90 s, as described in our previous report [15]. The treated starch suspension was dried at 40 °C for 24 h and microwave modified starch (MC) was obtained.

#### 2.2.3. Ultrasonic-Microwave and Microwave-Ultrasonic Samples Preparation

The obtained UC as mentioned in Section 2.2.1 was further treated by microwave as described in Section 2.2.2 to obtain ultrasonic microwave modified starch (UMC). 

Similarly, the MC as described in Section 2.2.2 was re-modified using the same ultrasonic procedure as described in Section 2.2.1 to obtain microwave-ultrasonic modified starch (MUC).

### 2.3. Morphological Structure of Chestnut Starch Granules

The morphology of the starch samples was observed by scanning electron microscopy (SEM) (JSM-6700F, JEOL Ltd., Tokyo, Japan). A small amount of each sample was evenly coated on dual-sided tape and subjected to gold plating. The accelerating voltage was 10.0 kV [14].

The starch sample was mixed with a 1:1 glycerin solution (glycerin/H_2_O, *v*/*v*) to make a 1% starch suspension. A drop of the starch suspension was observed by polarizing light microscope (PLM) (ZEISS imager. A2m; Carl Zeiss, Oberkochen, Germany), and images were taken at 25× magnification [16].

### 2.4. X-ray Diffraction (XRD)

The XRD patterns of starches were determined by Bruker D8 ADVANCE X-ray diffractometry (Bruker Corporation, Karlsruhe, Germany) at 40 kV and 40 mA. The scanning area was 5° to 45°, with a scanning rate of 2° /min, and a step size of 0.02° as a function of 2θ. The relative crystallinity of the chestnut starch was calculated by equation [17]:Crystallinity (%) = (Area of diffraction peak)/(Total area) × 100(1)

### 2.5. Fourier Transform Infrared (FTIR) Spectroscopy Analysis

The FTIR spectra of starches were recorded using a FTIR spectrometer (TENSOR II; Bruker Corporation, Ettlingen, German). The chestnut starch sample was thoroughly ground with dry KBr powder (1:100, *w*/*w*) and pressed in a mold to obtain tablets, and KBr was calibrated as a blank. The infrared spectra were obtained (32 scans) with a scan range of 4000 to 400 cm^−1^ at a resolution of 4 cm^−1^ [18].

### 2.6. Thermal Properties

The thermal properties of starches were studied using the method of Yang et al. [3] on a Q2000 Differential Scanning Calorimeter (DSC) (T.A. America Corporation, Osaka, Japan). The starch sample (3 mg) and distilled water (9 µL) were sealed in an aluminum pan overnight before heating over a range of 20–120 °C at 10 °C/min. The onset (*T_o_*), peak (*T_p_*), and conclusion (*T_c_*) temperatures, and gelatinization enthalpy (ΔH) were obtained.

### 2.7. Pasting Properties

The pasting properties of the samples were analyzed following the method of Joshi et al. [19] with some modifications. A 6% (*w*/*w*) starch slurry with a total weight of 28 g was placed in a TecMaster Rapid Viscosity Analyzer (RVA) (Perten Instruments, Hägersten, Sweden) to obtain a viscosity curve. To do this, the starch slurry was held at 50 °C for 1 min, heated from 50 to 95 °C at 12 °C/min, and held at 95 °C for 2.5 min, then cooled to 50 °C at the same rate and finally maintained at 50 °C for 2 min. Stirring was performed at 960 rpm for the first 10 s and then maintained at 160 rpm during operation. The measured parameters were peak, trough, final, breakdown and setback viscosities, and pasting temperature.

### 2.8. Swelling Power (SP)

A 0.2 g (*W*) starch sample was mixed with 10 mL of distilled water and heated in a water bath at 55, 65, 75, 85, or 95 °C for 30 min, cooled to room temperature, and then centrifuged at 12,000× *g* for 15 min. The supernatant was dried in an oven at 105 °C until reaching constant weight (*A*), and the weight of the swollen starch in the centrifuge tube was expressed as *P*. SP was calculated according to the following equations [20].
SP (g/g) = *P*/(*W* − *A*)(2)

### 2.9. Freeze-Thaw Stability

The freeze-thaw stability of starch can be measured by syneresis. Lower syneresis indicates better freeze-thaw stability. The 5% starch suspension was heated in a 95 °C water bath for 30 min with constant stirring. The paste was transferred to a pre-weighed centrifuge tube to determine the weight of the paste. The samples were subjected to alternating freezing and thawing cycles (freezing at −20 °C for 22 h, then thawing at 30 °C for 2 h) for 5 days, followed by centrifugation at 8000× *g* for 10 min after each cycle. Syneresis was calculated according to the following equation [21].
Syneresis (%) = (*M*_1_ − *M*_2_)/*M*_1_ × 100(3)
where *M*_1_ is the mass of the starch paste (g) and *M*_2_ is the mass of the starch precipitate after centrifugation (g).

### 2.10. Water Absorption Capacity (WAC) and Oil Absorption Capacity (OAC)

The WAC and OAC were analyzed as described in Nawaz et al. [9] with some modifications. A 0.5 g (*W*_0_) of starch powder was dispersed in 5 mL deionized water (for WAC) or canola oil (for OAC) in a centrifuge tube and weighed (*W*_1_). The mixture was vortexed for 30 min and then centrifuged at 4500× *g* for 10 min at room temperature. The supernatant was decanted, and the precipitate was weighed (*W*_2_). The WAC or OAC of starch was then calculated by the following equation.
WAC or OAC = [(*W*_2_ − *W*_1_)/*W*_0_](4)

### 2.11. Statistical Analysis

All analyses were carried out in triplicate. Analysis of the variance (ANOVA) of the experimental data was performed by SPSS 23.0 software (IBM Corporation, Armonk, NY, USA) with a significance level of 95% (*p* < 0.05). 

## 3. Results and Discussion

### 3.1. Morphological Structure

The microstructure of the native and modified chestnut starch samples was observed by SEM. The native chestnut starch (NC) exhibited a variety of shapes, ranging from round and oval, to the horn and irregular shapes (Figure 1), and the granules exhibited a smooth surface with flat edges, mostly large particles and some small particles, consistent with previous research results [2]. Deep pitting and cracks of particles were noticed during the ultrasonic treatment, and the surface of the starch subjected to microwave treatment displayed wrinkles and roughness. During the ultrasound process, the rapid formation and rupture of bubbles generated by cavitation produced high shear, inducing damage to the chestnut starch granules. At the same time, the outer layer of starch gradually detached from the periphery, which also promoted the formation and collapse of cavitation bubbles [3]. These results agreed with the findings of Kaur and Gill [18], who reported that ultrasonic-treated wheat, rice, maize, and barley starch particles appeared concave and cracked on the surface. During microwave treatment, pressure formed in the particles and caused the particles to expand, but the hydration of the particles cannot keep up with the expansion of the particles, causing the particles to collapse or even rupture [22]. In addition, the surface of UMC and MUC granules was more severely damaged where the granules showed rougher and more pits, indicating that the ultrasonic and microwave treatment promoted more drastic changes of the dual modified particles compared to single treatment. The microstructure of UMC particles differed from that of MUC particles which exhibited a relatively agglomerated state. This may be attributed to the greater sensitivity of the starch particles to microwave treatment [22], and the instantaneous high pressure and heat generated by the rapid collapse of the bubbles in the subsequent ultrasonic treatment caused the particles to aggregate. Deka and Sit [21] found a high agglomeration/fusion of dual-treated taro starch granules after microwave treatment, with more severe particle damage than that resulting from a single treatment.

The crystal structure of the starch granules was observed by PLM, and the polarized micrographs of chestnut starches are shown in Figure 1. Native starch granules have a semi-crystalline structure, and the crystalline structure and the amorphous structure exhibited anisotropy in density and refractive index under a polarizing microscope, resulting in birefringence and exhibiting a unique characteristic of the Maltese cross [23]. The change in the appearance of the Maltese cross reflected the change in the crystal structure of the internal particles [16]. Ultrasonic treatment altered the luminance of the chestnut starch Maltese cross, and the black areas spread, which may be due to the disrupting of the crystal layer of starch granules by ultrasound. The MC showed blurry crosses, and this was attributed to the destruction of the ordered arrangement of the crystalline regions under microwave treatment [23]. In addition, the cross luminance of MUC was observed to fade and some crosses disappeared, and the UMC granules showed weak birefringence with no obvious Maltese cross, indicating that ultrasonic and microwave dual treatment severely damaged the double helices structure of amylopectin in the crystalline thin layer of starch [24].

### 3.2. Long- and Short-Range Molecular Order

The crystallinity of starch reflects its long-range molecular order. Analysis of NC revealed a typical C-type crystal with strong characteristic peaks at 5.6°, 15°, 17°, and 23° (2θ) (Figure 2A), which is consistent with a previous report [14]. Compared to NC, the intensity of the diffraction peaks at 5.6°, 15°, 17°, and 23° (2θ) were weakened in the modified starches, but the crystallinity pattern of the samples showed little change. Yang et al. [3] also observed that ultrasonic treatment had little effect on the crystal form of waxy corn starch. The results of the present study showed that the ultrasonic and microwave modification resulted in little alteration of the crystal polymorph of the starch granules. The relative crystallinity of NC was 27.85%, which was higher than that of the modified chestnut starches (Figure 2A). The MUC showed the lowest relative crystallinity (23.75%), followed by UMC (23.98%). Deka and Sit [21] found that dual modification of microwaves and other heat treatment methods reduced the relative crystallinity of taro starch, and pointed out that the weakening in the peak intensity might be caused by the loss of the crystal region due to hydrogen bond cleavage. The decrease in the relative crystallinity could be attributed to the destabilization of the layered arrangement of starch, and the double helix structure of the amylopectin crystallized region became more fragile under ultrasound and microwave conditions [13,25], and this decrease was more obvious under the dual treatment. A decrease in relative crystallinity has also been reported in microwave and heat moisture treated rice starch by Guo et al. [26] and waxy corn starch sonicated by Yang et al. [3].

The FTIR spectra of native and modified chestnut starches are shown in Figure 2B, and the intensity ratio of 1047/1022 cm^−1^ indicates the short-range order of the inner granules of the starch molecule [18]. The characteristic bands of the FTIR spectra could be divided into the following four regions: 3600–3000 cm^−1^ (O-H stretching region), 3000–2800 cm^−1^ (C–H stretching regions), 1500–800 cm^−1^ (the fingerprint region), and below 800 cm^−1^ [27]. As shown in Figure 2B, no new absorption peaks and no loss of characteristic absorption peaks in all modified starches suggested no loss or gain of functional groups, which indicated ultrasound and microwave treatments only exhibited physical effects on the starch samples [28]. Compared with the NC, the 1047/1022 cm^−1^ of the MC increased by 0.1249, but the ratios of the other modified samples decreased (Table 1), while the UC decreased by 0.0202, and the UMC and MUC decreased by 0.0585 and 0.0107, respectively. Nawaz et al. reported that the crystalline order of microwave-treated lotus seed starch was higher than that of native starch [9], but Monroy et al. observed a decrease in the ordered structure of cassava starch after ultrasound modification [29]. These results indicated that ultrasonic treatment disrupted the association of starch chains and decreased the short-range molecular order [25], but microwave treatment promoted the molecular rearrangement of the starch crystalline region, promoting the formation of a relatively ordered structure [9]. For UMC samples, starch granules were attacked by ultrasonic treatment firstly, resulting in the severely damaged crystalline regions and ruptured starch chains. Therefore, the structure became loose and subsequent microwave treatment promoted this change, which was in accordance with the lowest 1047/1022 cm^−1^ ratio of UMC. In the MUC sample, the preferential treatment of microwaves promoted the rearrangement of molecules, which led to the weakening of the synergistic effect of subsequent ultrasonic treatment.

### 3.3. Thermal Properties

The thermal properties of chestnut starches modified by ultrasonic and microwave treatments are shown in Table 1. Among all starch samples, UC had the lowest *T_o_* (57.5 °C) and *T_p_* (62.2 °C), while UMC had the highest *T_o_* (61.8 °C) and *T_p_* (64.3 °C). The *T_c_* of all modified samples was reduced compared to NC (68.0 °C). *T_o_* and *T_c_* are related to the melting temperatures of the weakest crystallites and high-perfection crystallites in the starch granules, respectively [30]. The increase in *T_o_* might reflect the partial melting of the weakest crystallite of the particles becoming more stable and structural changes within the particles affected by microwave heating [31]. Previous studies reported similar trends in rice subjected to microwave radiation [32]. The decrease in *T_c_* was attributed to the formation of unstable crystallites by the modification treatment [33]. The difference in the gelatinization temperature of the modified starches might be due to the tightness of the particle structure after treatment [19].

The ΔH of NC was 2.91 J/g and consistent with the results of Guo et al. (0.65–5.52 J/g) [1]. Ultrasonic and microwave modification treatments reduced the ΔH of all single- and dual-treated samples, and the UMC had the lowest ΔH of 0.97 J/g, followed by MUC of 1.18 J/g. There was a significant positive correlation between ΔH and the relative crystallinity (*r* = 0.824, *p* < 0.01). The decrease in ΔH demonstrated the destruction of the double helices structure under ultrasonic or microwave treatment [34], which was consistent with the results of PLM. A decrease in ΔH was also observed in ultrasonic-treated cassava [29], and microwave radiation-treated Indian Horse chestnut starch [35].

### 3.4. Pasting Properties of Chestnut Starches

The pasting properties of native, ultrasonic-, and microwave-modified chestnut starches determined by RVA are shown in Figure 2C, with the parameters summarized in Table 1. The pasting temperature was 72.7 °C in NC, higher than those in the modified samples. The decrease in the pasting temperature of all modified chestnut starches indicated that the ultrasonic and microwave treatments reduced the resistance of the starch to swelling [3]. The pasting properties including the peak, trough, final, and breakdown viscosities of chestnut starch were reduced after ultrasonic and microwave modification, which might be due to the degradation of starch molecules induced by ultrasonic or microwave treatment. The MC and MUC showed a low peak, trough, and breakdown viscosity, which may be related to the degradation of glycosidic bonds promoted by the vibration of polar molecules during microwave treatment. The peak viscosity showed a significant correlation with relative crystallinity (*r* = 0.818, *p* < 0.01) and ΔH (*r* = 0.624, *p* < 0.05). Ultrasound and microwave treatment could disrupt the crystallinity structure of starch granules, rupture the macromolecular chains, and reduce the rigidity and integrity of the granules, thus reducing the peak viscosity [13]. The breakdown viscosity measures the extent of disintegration of the starch granules, and the lower breakdown viscosity suggested the stronger resistance of the starch granules to shear-thinning during cooking [36]. The breakdown viscosity of UC, MC, UMC, and MUC samples were found to be 938, 632, 1246, and 406 cP, respectively. Therefore, the MUC sample was the most resistant to shear thinning during cooking. The breakdown viscosity of UMC was the highest among the modified samples, which indicated that the physical structure of the particle was weaker, making it more likely to collapse under heat and shear treatment [37]. A similar trend was observed in ultrasonic-treated waxy corn starch [3], quinoa flour [8], and taro starch treated by microwave and then autoclave treatment [21]. The setback viscosity reflects the retrogradation capacity of starchy foods [35]. Analysis of the setback viscosity found that the UC sample decreased slightly, and the MC increased compared with NC. This phenomenon may be due to the interference of ultrasonic amylose rearrangement, while microwave treatment played a positive role [22]. The setback viscosity of UMC and MUC were between MC and UC, and higher than that of NC, indicating the neutralization of the second modification. In addition, dual treatment accelerated the aging of starch.

### 3.5. Swelling Power

NC showed the highest SP (31.83 g/g) at 95 °C (Table 2). Compared with NC, the SP of the modified starches reduced when reaching the temperature of 75 °C. The high SP is due to the high proportion of long-chain amylopectin in starch [21]. The UMC presented the lowest SP (20.54 g/g) at 95 °C, followed by MUC (22.81 g/g). Both single and dual treatment reduced the SP of NC. The decrease in SP at high temperature can be attributed to the destruction of the granular structure of the starch by ultrasonic and microwave treatments, resulting in the degradation of the amylopectin chain. Furthermore, starch is gelatinized at high temperature, and the gelatinized starch and denatured protein matrix can prevent the diffusion of water into the starch matrix [36]. Microwave irradiation induced the swelling of the restricted particles due to recombination within the particles [38]. Ding et al. pointed out that ultrasonic treatment reduced the swelling power of retrograded starch [25]. In addition, the SP showed a significant correlation with pasting temperature (*r* = 0.716, *p* < 0.01) and peak viscosity (*r* = 0.701, *p* < 0.01). The structures of dual-modified starches (UMC and MUC) at a high temperature were more seriously damaged than those subjected to a single treatment (UC and MC), which restricted the starch swelling due to the fewer starch crystallites, resulting in a lower SP.

### 3.6. Freeze-Thaw Stability

The freeze-thaw stability reflects the degree of syneresis after the thawing of frozen foods, where the lower the syneresis rate, the better the freeze-thaw stability. The syneresis of chestnut starches increased with the number of freeze-thaw cycles (Table 3). After 5 cycles, maximum syneresis occurred in NC (11.59%) and minimum in MUC (1.00%), and the syneresis of chestnut starches decreased significantly (*p* < 0.05) after the single (UC and MC) and dual treatments (UMC and MUC). The freeze-thawed starch gel formed a spongy structure, and the treatment might improve the stability of the gel by delaying the formation of this spongy network [39]. Additionally, ultrasonic or microwave treatment caused the cleavage of the starch molecular chain and the destruction of the hydrogen bonding force between amylose–amylose and amylose–amylopectin [36]. Consequently, the syneresis was lower in modified samples (UC, MC, UMC and MUC) than in NC.

### 3.7. Water (WAC) and Oil (OAC) Absorption Capacities 

The WAC of chestnut starches ranged from 1.05 to 2.37 g/g (Table 3). The WAC of modified starches, except for the UC sample, significantly increased (*p* < 0.05) after ultrasonic and microwave treatment. The highest WAC was observed in MUC (2.37 g/g), followed by UMC (2.14 g/g). Ultrasonic and microwave treatments caused degradation of starch to dextrin, maltose, and glucose. These monosaccharides had a higher affinity for water than starch, thus increasing the water absorption capacity of starch [40]. In addition, the results showed that the WAC was significantly negatively correlated with the relative crystallinity (*r* = –0.939, *p* < 0.01) and ΔH value (*r* = –0.932, *p* < 0.01). Alimi and Workneh [41] also found that WAC was inversely proportional to the relative crystallinity. 

The OAC of the modified chestnut starches was higher than that of NC (1.23 g/g), with UMC exhibiting the highest value (1.69 g/g). An increase in OAC might be due to the destruction of the starch structure by ultrasonic and microwave treatments, resulting in more hydrophobic portions of the amylose–lipid complex being exposed [42]. Similarly, microwave-treated lotus seed starch exhibited water and oil absorption capacities that were higher than those of native starch [9].

## 4. Conclusions

After single ultrasonic or microwave treatment and dual-modification on chestnut starch, UMC and MUC showed lower SP, relative crystallinity, and ΔH than UC and MC, due to the synergistic effect of ultrasonic and microwave treatment, and the layered structure and double helix structure of starch was more severely disrupted. This was confirmed by the observation of more wrinkles and grooves on the particle surfaces and fewer Maltese crosses of dually modified starch. The pasting properties of all modified starches decreased, whereas the freeze-thaw stability was better than that of the native chestnut starch, suggesting its potential application in frozen foods. In addition, the highest WAC was observed in MUC, and the UMC showed the largest OAC. The results of this study can provide theoretical support for the dual physical modification of starch, and provide a new reference for the industrial production of starch-based products that require specific properties, e.g., starches with high water absorption can be used as thickeners.

## Figures and Tables

**Figure 1 polymers-12-01718-f001:**
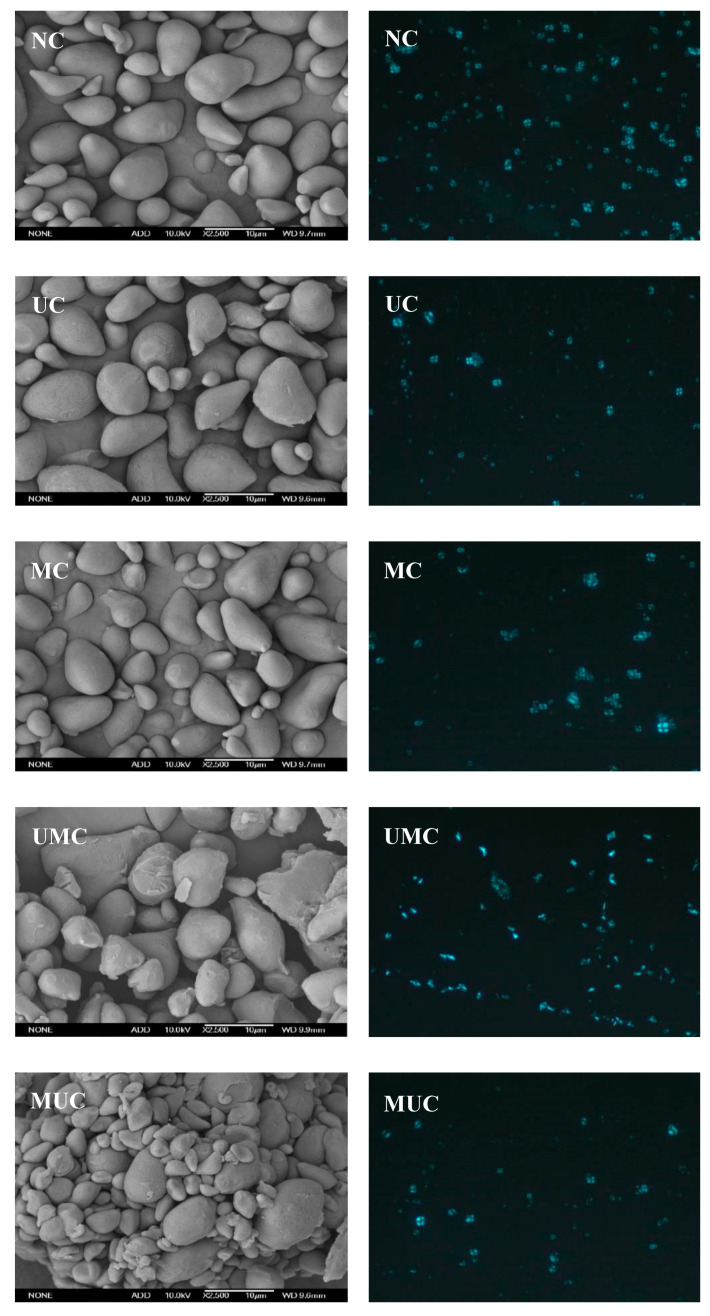
Scanning electron microscopy (2500×) (**left**) and polarized light microscopy (25×) (**right**) of native and modified chestnut starches. NC: native chestnut starch, UC: ultrasonic modified starch, MC: microwave modified starch, UMC: ultrasonic-microwave modified starch, MUC: microwave-ultrasonic modified starch.

**Figure 2 polymers-12-01718-f002:**
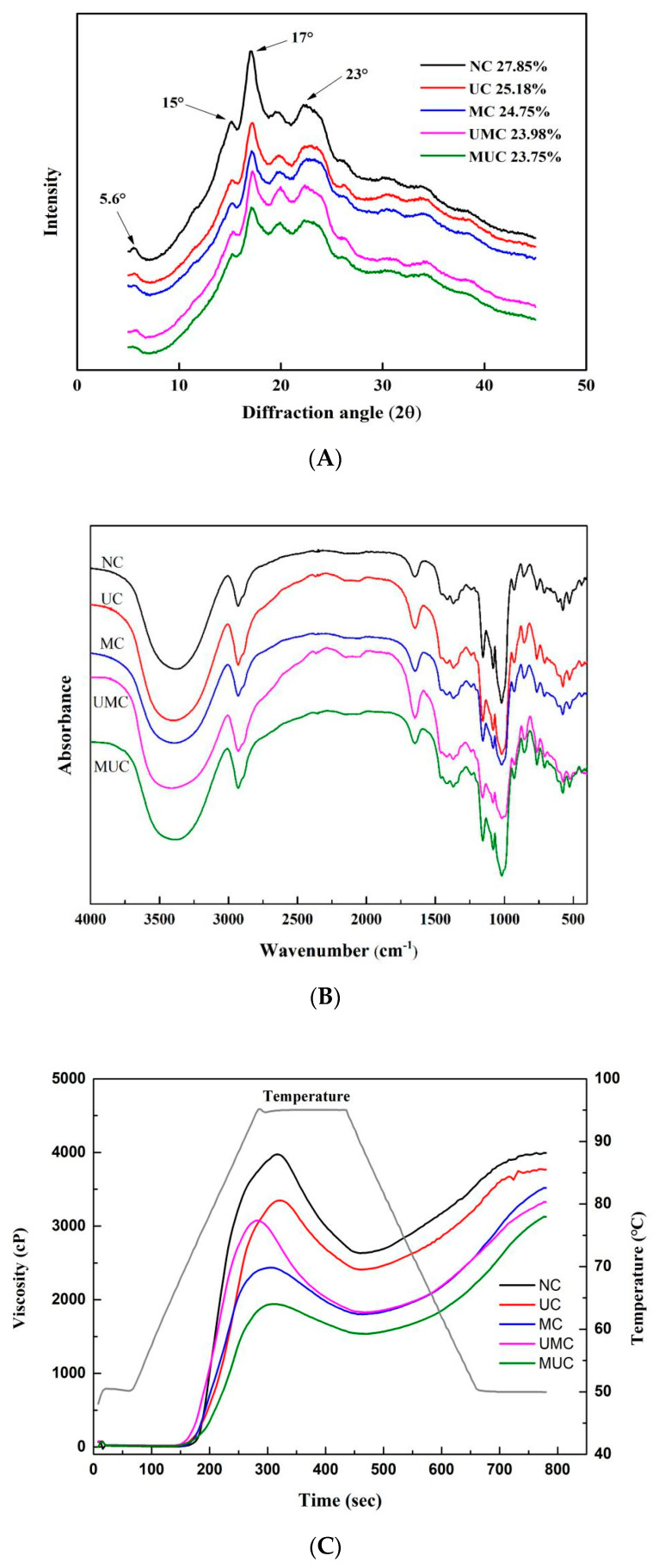
X-ray diffraction diagrams (**A**), FTIR spectra (**B**) and Rapid Viscosity Analyzer (RVA) pasting profiles (**C**) of native and modified chestnut starches. The values mean relative crystallinity (**A**). NC: native chestnut starch, UC: ultrasonic modified starch, MC: microwave modified starch, UMC: ultrasonic-microwave modified starch, MUC: microwave-ultrasonic modified starch.

**Table 1 polymers-12-01718-t001:** Thermal, FTIR spectra and pasting parameters of native and modified chestnut starches.

Samples	*T_o_* (°C)	*T_p_* (°C)	*T_c_* (°C)	ΔH (J/g)	IR Ratio of 1047/1022 cm^−1^
NC	59.8	64.2	68.0	2.91	1.1319
UC	57.5	62.2	66.6	2.61	1.1117
MC	59.8	64.0	67.8	2.30	1.2568
UMC	61.8	64.3	67.2	0.97	1.0734
MUC	59.9	63.1	66.8	1.18	1.1212

The data are presented as average values. *T_o_*, onset temperature; *T_p_*, peak temperature; *T_c_*, conclusion temperature; ΔH, endothermic enthalpy of gelatinization. NC: native chestnut starch, UC: ultrasonic modified starch, MC: microwave modified starch, UMC: ultrasonic-microwave modified starch, MUC: microwave-ultrasonic modified starch.

**Table 2 polymers-12-01718-t002:** Swelling power (g/g) of native and modified chestnut starches.

Samples	55 °C	65 °C	75 °C	85 °C	95 °C
NC	2.28 ± 0.33 ^a,b^	7.31 ± 0.12 ^a^	13.08 ± 0.13 ^a^	18.09 ± 1.44 ^a^	31.83 ± 0.14 ^a^
UC	2.32 ± 0.11 ^b^	7.99 ± 0.04 ^a^	11.75 ± 0.02 ^b^	15.76 ± 1.47 ^b^	26.33 ± 0.42 ^b^
MC	2.74 ± 0.25 ^a^	8.43 ± 0.57 ^a^	11.21 ± 0.04 ^b,c^	14.88 ± 0.53 ^b^	24.31± 0.13 ^c^
UMC	2.52 ± 0.05 ^a^	7.46 ± 0.44 ^a^	9.65 ± 0.34 ^d^	15.57 ± 0.01 ^b^	20.54 ± 0.14 ^d^
MUC	2.86 ± 0.05 ^a^	7.82 ± 0.19 ^a^	10.36 ± 0.15 ^c,d^	18.15 ± 0.03 ^a^	22.81 ± 0.51 ^c^

The data are presented as mean values ± SD (*n* = 3). The different letters for values in the same column indicate significant differences (*p* < 0.05); NC: native chestnut starch, UC: ultrasonic modified starch, MC: microwave modified starch, UMC: ultrasonic-microwave modified starch, MUC: microwave-ultrasonic modified starch.

**Table 3 polymers-12-01718-t003:** Syneresis, water (WAC) and oil (OAC) absorption capacities of native and modified chestnut starches.

Samples	Syneresis (%)					WAC (g/g)	OAC (g/g)
	Cycle 1	Cycle 2	Cycle 3	Cycle 4	Cycle 5		
NC	0.72 ± 0.04 ^a^	2.17 ± 0.10 ^a^	3.81 ± 0.13 ^a^	8.27 ± 0.02 ^a^	11.59 ± 0.17 ^a^	1.05 ± 0.15 ^d^	1.23 ± 0.04 ^b^
UC	0.42 ± 0.03 ^b^	0.51 ± 0.06 ^b^	1.13 ± 0.05 ^b^	2.48 ± 0.17 ^b^	3.66 ± 0.10 ^b^	1.46 ± 0.03 ^c,d^	1.40 ± 0.04 ^a,b^
MC	0.63 ± 0.06 ^a^	0.82 ± 0.09 ^b^	1.02 ± 0.04 ^b^	1.33 ± 0.04 ^c^	1.67 ± 0.03 ^c^	1.76 ± 0.09 ^b,c^	1.57 ± 0.17 ^a,b^
UMC	0.17 ± 0.01 ^c^	0.70 ± 0.06 ^b^	0.98 ± 0.05 ^b^	1.35 ± 0.10 ^c^	1.91 ± 0.08 ^c^	2.14 ± 0.05 ^a,b^	1.69 ± 0.14 ^a^
MUC	0.68 ± 0.02 ^a^	0.79 ± 0.02 ^b^	0.88 ± 0.04 ^b^	0.97 ± 0.01 ^c^	1.00 ± 0.03 ^d^	2.37 ± 0.01 ^a^	1.37 ± 0.13 ^a,b^

The data are presented as mean values ± SD (*n* = 3). The different letters for values in the same column indicate significant differences (*p* < 0.05); NC: native chestnut starch, UC: ultrasonic modified starch, MC: microwave modified starch, UMC: ultrasonic-microwave modified starch, MUC: microwave-ultrasonic modified starch.

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
