# Peer review of "Effect of Ultrasonic and Microwave Dual-Treatment on the Physicochemical Properties of Chestnut Starch"

_polymers, 2020, doi:10.3390/polym12081718_

Round 1
Reviewer 1 Report
line 51 - Previous reports also pointed out that dual physical modification is more effective than single treatment. - references required for confirmation, not only self-citation.
line 55 - References required to confirm few studies on the simultaneous effects of ultrasound and microwave modification on chestnut starch
line 76 "The preparation of the microwave starch suspension was the same as that of the ultrasonically treated samples." The phrase is wrong, the procedure of MW treatment is totally different from US treatment.
line 184 Use the same acronyms in the figure and in the legend to the figure. For example, MU - in the figure, but MUC for microwave-ultrasonic modified starch - in the legend.
The study is of low scientific interest. There are plenty of starchy botanic species, and starch from many of them was subjected to ultrasound and microwave treatment with the same or close results. So, all findings are quite secondary, mostly of technical, not fundamental character. Not for Q1 journal.
Reviewer 2 Report
The paper is quite interesting, fairly done, and could be published in Polymer. However, some relevant shortcoming makes this paper impossible to publish in the current form.
Introduction
In this part almost all necessary information are given. However, chestnut starch in not popular around all the world, and its properties are not well known within starch scientist. I suggest, to supplement Introduction by information regarding crucial differences in properties of chestnut starch as compared to the most popular commercial starches (including waxy corn).
Materials and methods
The description of the methods for ultrasonic and microwave treatment is confusing. First of all, it is unclear whether 2g refers to the amount of starch or the amount of the whole suspension. The mass of the material subjected to the treatment by particular portion of energy is of crucial importance.
Moreover, there are no information whether mixing was applied by microwave treatment.
Results and discussion
IR spectra:
Discussion of IR spectra seems to be to much arbitrary. The inference regarding splitting of glycosidic bond only on the basis of IR spectrum seems to be excessive.
DSC study:
Gelatinisation enthalpy of native chestnut starch is very low (several times smaller) as compare to gelatinisation enthalpy of the most common commercial native starches. Why? Is this typical for all chestnut starches or it is caused by isolation procedure?
Pasting characteristics:
There is no need (this is wrong practice) to give the data of pasting characteristics in the table 1. The figure 2 C presents these data in excellent form.
Conclusions
The beginning of the Conclusions parts seems to be more Summary than Conclusions.
However, more importantly, the last sentence (lines 341-344) is not supported by the results. There are no data regarding costs and environment effect of ultrasound and microwave treatment. Moreover, there are no data regarding applicability of that way modified starches in the industrial practice.
Round 2
Reviewer 1 Report
Authors did a good job adding some references in the introduction part. The paper is well written and experiments are performed on a high level. Nevertheless, the topic of the paper, material, methods and results are not that significant, at least , not for Q1 journal. Authors used well-known material (starch), well-known methods, and obtained changes in the material, that do not demonstrate any really new effects. So, the paper is about another example of starch treated with known procedures - in different variants. One may imagine a huge number of similar papers with starches of different botanic origin and applied different physical treatments. In every case there will be somewhat effect. In my opinion, Q1 journal should be interested in more innovative research, or novel findings, more original experiments. Good paper for Q3 journal.
Reviewer 2 Report
The Authors improved the original version of the manuscript. The current version can be published in current form. The only small change that could be made is given the information regarding of the mixing (or not) of the suspension during microwave treatment.